# Automated Science Scheduling for the ECOSTRESS Mission

**Amruta Yelamanchili, Steve Chien, Alan Moy, Elly Shao, Michael Trowbridge,**
**Kerry Cawse-Nicholson, Jordan Padams, Dana Freeborn**

Jet Propulsion Laboratory
California Institute of Technology
4800 Oak Grove Drive
Pasadena, CA 91109
{firstname.lastname}@jpl.caltech.edu

## Abstract

We describe the use of an automated scheduling system for observation policy design and to schedule operations of the NASA (National Aeronautics and Space Administration) ECOSystem Spaceborne Thermal Radiometer Experiment on Space Station (ECOSTRESS). We describe the adaptation of the Compressed Large-scale Activity Scheduler and Planner (CLASP) scheduling system to the ECOSTRESS scheduling problem, highlighting multiple use cases for automated scheduling and several challenges for the scheduling technology: handling long-term campaigns with changing information, Mass Storage Unit Ring Buffer operations challenges, and orbit uncertainty. The described scheduling system has been used for operations of the ECOSTRESS instrument since its nominal operations start July 2018 and is expected to operate until mission end in Summer 2019.

## Introduction

NASA's ECOSTRESS mission (NASA 2019) seeks to better understand how much water plants need and how they respond to stress. Two processes show how plants use water: transpiration and evaporation. Transpiration is the process of plants losing water through tiny pores in their leaves. Evaporation of water from the soil surrounding plants affects how much water the plants can use. ECOSTRESS measures the temperature of plants to understand combined evaporation and transpiration, known as evapotranspiration.

ECOSTRESS launched on June 29, 2018 to the ISS (International Space Station) on a Space-X Falcon 9 rocket as part of a resupply mission. The instrument is attached to the Japanese Experiment Module  Exposed Facility (JEM-EF) on the ISS and targets key biomes on the Earth's surface, as well as calibration/validation sites. Other science targets include cities and volcanoes. From the orbit of the Space Station (Figure 1), the instrument can see target regions at varying times throughout the day, rather than at a fixed time of day, allowing scientists to understand plant water use throughout the day.

The instrument used for ECOSTRESS is a thermal infrared radiometer. A double-sided scan mirror, rotating at a constant 25.4 rpm, allows the telescope to view a 53°-wide nadir cross-track swath with one scan per 1.18 seconds. The

nominal observation unit is a scene, made up of 44 scans, and takes roughly 52 seconds to acquire. For simplification of operations, we consider that ECOSTRESS scenes are 52 seconds long. About 1000 scenes may be acquired in a given week. Figure 2 shows a set of planned observations over North America. Each square represents one 52-second long scene.

CLASP (Knight and Chien 2006) was initially used pre-launch as a tool to analyze the addition of a new science campaign. CLASP was then used for operations to generate command sequences for the instrument. The command sequences are translated from the observation schedule generated by CLASP, and include other time and location dependent instrument actions besides observations, such as hardware power cycles through high radiation environments.

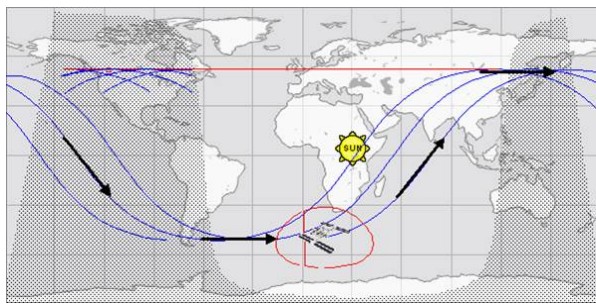

Figure 1: Three Orbital Tracks of the ISS (Robinson 2013)

Each mission comes with its own set of challenges, and there were three specifically that required adaptations to CLASP as follows.

- ECOSTRESS has a long-term science campaign that we need to satisfy. From week to week, the orbital ephemeris can change, and thus the schedule needs to be updated each week. We need to be able to account for previously executed observations when scheduling for the future.

- An issue with the instrument Mass Storage Unit (MSU) was discovered, and rather than performing an instrument firmware update, we proposed a ground-based solution that accounts for this additional complexity in the data modeling in the schedule.

- The uncertainty in the orbital ephemeris (predictions of

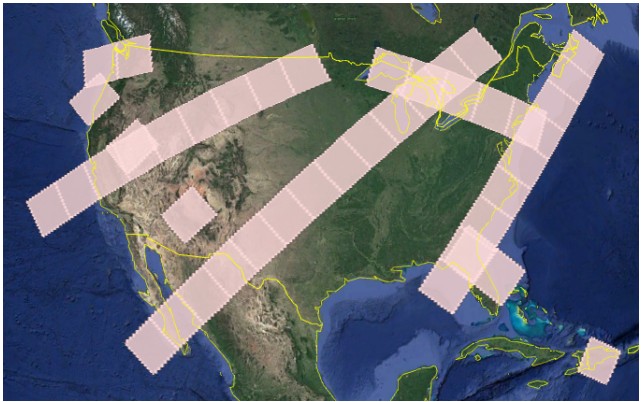

Figure 2: Observations Over North America

the spacecraft location) required scheduling additional observation time to ensure no targets are missed.

In the remainder of this paper, we describe these operational challenges and how we addressed them successfully. We also validate our methods used through computational analysis.

## Initial Scheduling Problem

CLASP recieves as input (Figure 3) the ephemeris of the ISS (predicted time-tagged locations), instrument constraints, and a set of set of science campaigns, which are made up of:

- target regions of interest on the Earth's surface (Figure 4)
- illumination constraints
- a priority

We want to produce an observation schedule to view these regions as many times as possible while respecting constraints such as memory capacity, downlink rate, and keepout zones (e.g. high radiation environments) where we do not want to take any observations. Science campaigns can be target regions or single point locations. We generate a gridded approximation of target regions for faster computation, to get a set of target points.

CLASP uses the CSPICE Toolkit provided by the Navigation and Ancillary Facility (NAIF) (Acton 1996) at JPL to do geometric reasoning regarding the visibility swaths of instruments from the spacecraft they are attached to. The size, shape, and location of the swaths depend on the position and orientation of the spacecraft, and the field-of-view of the instrument. CLASP has the capability to schedule instruments that can point off-nadir, but ECOSTRESS specifically has no pointing capability, so each observation spans the whole range of its field-of-view. The planning horizon is broken up into fixed duration observations, and CLASP computes the intersection between the target grid points and the observations. We use a one-pass greedy scheduling algorithm to place observations according to the priority of the targets they cover.

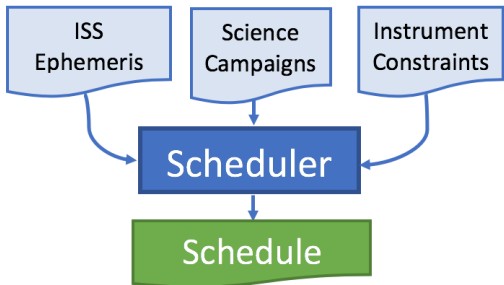

Figure 3: Scheduler inputs and outputs

## Problem Statement

The CLASP problem statement (Knight and Chien 2006):
  Given:

- a set of regions of interest $R = \{r_1, ..., r_n\}$
- a temporal knowledge horizon $(hst, het)$ over which we know the vehicle's activities
- a set of observation opportunities $O = \{o_1, ..., o_n\}$ within the horizon $(hst, het)$ where each $o_i \in O$ consists of a start $(o.start)$ and a duration $(o.duration)$
- a set of instrument swaths $I = \{i_1, ..., i_n\}$ where $\forall (o_i \in O) \exists (r_i, i_i) \mid (\text{grid}(o_i) \in \text{grid}(r_i)) \wedge (\text{grid}(o_i) \in \text{grid}(i_i))$
- a scoring function $U(r_i)$
- keepout zones where observations should not be taken
- a bound on memory $M_{max}$
- a rate at which memory is used while the instrument is on $\dot{M}_{fill}$
- a rate at which memory is recovered during downlink $\dot{M}_{drain}$, which occurs when the instrument is not observing and is not in a keepout zone

Our goal is to select $A \subseteq O$ to maximize $U(r_i) \, \forall r_i \in R$ subject to instrument constraints involving available memory and keepout zones.

We introduce new and modified ECOSTRESS-specific components to the problem statement in subsequent sections.

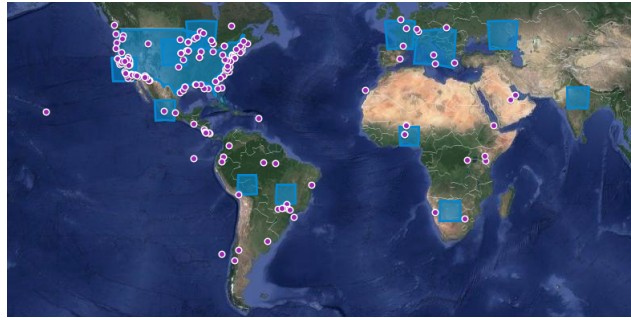

Figure 4: Science Campaigns can be made up of regions (blue boxes) or point targets (purple dots)

## Sliding Window Scheduling and Changing Ephemeris

If we had perfect knowledge of the future and a perfect observer, we could simply schedule the entire mission at once and feed the instrument slices of the schedule to execute. Unfortunately, there are uncertainties that require the schedule to be updated:

- new or modified science campaigns
- special maneuvers for spacecraft docking that could put the instrument in an unsafe position
- thruster burns to counteract orbital decay that change the trajectory

The goal for most of the ECOSTRESS science campaigns is to observe them whenever possible in daylight, to see how their water use changes at different times throughout the day over an extended period of time. During pre-operations analysis, CLASP was used to understand how to effectively make use of unused data volume. A new campaign was thus inserted into the ECOSTRESS science operation goals - to construct daytime maps of the global landmass. We found that attempting to construct one global map per month would not violate any instrument memory constraints, and allow all of the primary science campaigns to still be fulfilled as much as possible.

The ECOSTRESS payload is commanded weekly after a new ISS ephemeris prediction is received, uploading two weeks worth of command sequences to the instrument. Operationally, only the first week of sequences will get executed before the next set of sequences is uploaded, with the second week only being executed if, for some reason, a new schedule is not able to be uploaded the next week. Since the global map takes four weeks to construct, but only two weeks worth of sequences are planned, we need to determine which parts of the global landmass have been previously observed, so in the next schedule we can attempt to observe currently unobserved regions. This required adapting CLASP to be able to receive a previous schedule as input, and account for those observations in scheduling for the current planning horizon.

Time is divided into three regions with varying certainty: past (high certainty), current (moderate certainty) and future (low certainty). All three regions contribute to the score of a schedule against the science observation campaigns. We force a boundary condition that the data recorder is empty at the end of each planning period to simplify operations. We add the following component to our problem statement:

- **a planning horizon $(\text{phst}, \text{phet}) \subseteq (\text{hst}, \text{het})$ over which the schedule may be modified**

The schedule is a living document that is updated weekly during operations. Figure 6 shows the inputs and outputs of the scheduler for each week's run.

## Ring Buffer Scheduling Constraint

The MSU onboard ECOSTRESS is a ring buffer. Ring buffers consist of two pointers - a read pointer $(r(t))$ and a write pointer $(w(t))$ (Knuth 1997). During downlink, the

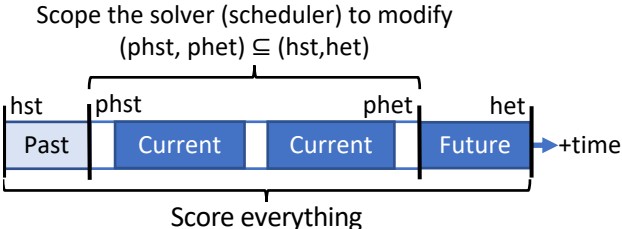

Scope the solver (scheduler) to modify
(phst, phet) ⊆ (hst,het)

Figure 5: Planning horizon accounting for the past

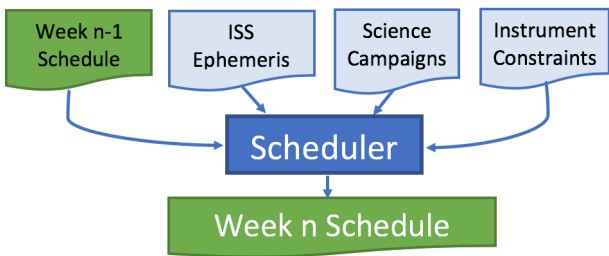

Figure 6: Scheduler now takes in previous week's schedule

data is read from the read pointer position, and the read pointer advances. New data is written to the write pointer position, and the write pointer advances. When functioning correctly, the read and write pointers move back to the start of memory when they reach the end of memory. An issue in the instrument firmware causes the read pointer to stay at the end of the memory rather than move to the start of memory as expected, continuously reading the same data from that position, even though new data may have been written at the start of memory.

The undesired condition resulting in data loss occurs when

$$w(t) < r(t) \tag{1}$$

indicating the write pointer has wrapped back around but the read pointer has not.

Rather than update the instrument firmware, which poses a higher risk, we opted to attempt a ground-based solution. A command can be issued that will reset the pointers back to the start. When scheduling the pointer reset times as well as the observations, we consider two constraints:

- **Constraint 1**: The amount of data acquired in between reset commands should not exceed the capacity of the buffer.
- **Constraint 2**: At the time of a reset command, the locations of the read pointer and the write pointer should be equal.

Constraint 1 prevents the write pointer from wrapping around the buffer, and Constraint 2 prevents any undownlinked data from being in the buffer at the time of a reset. If either constraint is not met, data will be lost.

Both constraints are specific to the ECOSTRESS mission and does not apply to the CLASP problem in general. Our scheduling goal then changes to:

- Our goal is to select $A \subseteq O$ to maximize $U(r_i) \ \forall r_i \in R$ subject to instrument constraints involving available memory, keepout zones, and Constraints 1 and 2.

We schedule in two passes, outlined in Algorithm 1. The first pass determines the ring buffer reset times, and the second pass returns the final schedule. In the initial pass, the scheduler is run with just the highest priority targets, with reset times at the end of each week. New schedules are uploaded weekly, so having the buffer empty at the end of each week allows for a more simple handover. When scheduling the observations, CLASP enforces the above constraints. We then examine the memory profile of the resulting schedule. We search forward through the memory profile until the point in time when the data has filled to some fraction of the buffer. This fraction is an estimate of the amount of memory going towards the high priority data, so there is enough memory still available to observe lower priority targets. Moving backwards from this point, we look for a time when the amount of memory onboard is lower than some threshold. If there is no memory onboard at a specific time, that means we are able to place a ring buffer reset there without sacrificing observing an high priority targets for that time period in the final schedule. The larger the amount of data scheduled to be in the buffer, the more observations will fail to be scheduled in the next run of CLASP. If a suitable point is not found, the threshold increases and the process repeats until a time for the reset is found. Then the search continues moving forward from the time chosen for the reset, and this repeats until we reach the end of the planning horizon. Figure 7 shows an example of reset times chosen after examining the memory profile.

Once all the reset times are found, the scheduler is run again with high and low priority targets to produce the final schedule, enforcing Constraints 1 and 2. Figure 8 shows the memory profile with data from high and low priority campaigns, and the buffer is empty at the reset times.

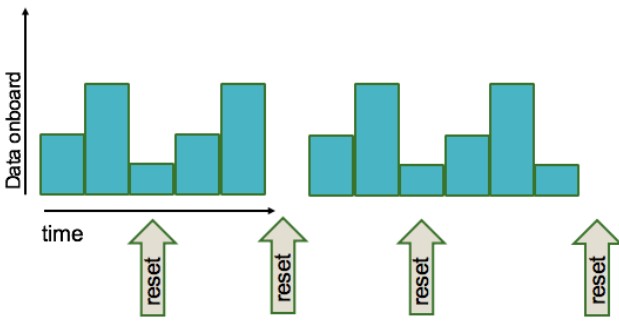

Figure 7: Scheduling resets after looking at memory profile with data from high priority campaigns (blue)

## Uncertainty of Predicted Ephemeris

The ISS is in a region of orbit known as Low Earth Orbit (LEO). Objects in LEO experience drag from the atmosphere, which results in the ISS experiencing some drift

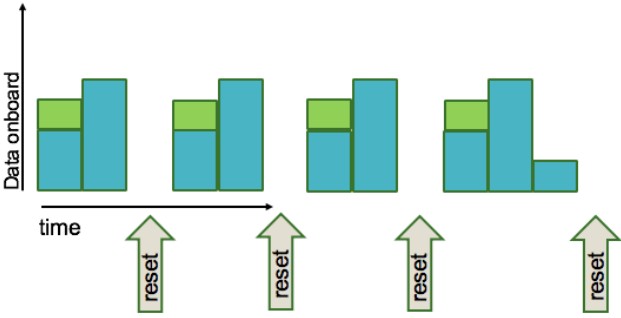

Figure 8: Memory profile of final schedule with memory from high priority campaigns (blue) and low priority campaigns (green), with no data in the buffer when resets occur

from its predicted location. This can cause an observation to be taken that misses the region it was intended to observe.

In the original version of CLASP, each observation has a start time ($o.start$), and a duration ($o.duration$). For ECOSTRESS, the duration is fixed at 52 seconds long. If a target is predicted to be viewed at any time between $o.start$ and $o.start + o.duration$, that target is satisfied by that observation. If a target was predicted to be viewed near the start or the end or the observation window, that target may be missed operationally due to the uncertainty in the ISS position.

The initial solution to this problem was to spend extra time observing before and after each set of contiguous observations. Because the ECOSTRESS instrument takes data in scenes lasting 52 seconds, we add 26 seconds of observational time before and after, so each set of contiguous observations still has a duration that is a multiple of an ECOSTRESS scene. During scheduling, this extra time is accounted for when checking data volume constraints, but those times are not considered to satisfy any science targets. However, this extra time spent observing is wasteful and takes up data volume that could potentially be used by other productive observations.

A solution that could schedule observations such that no science targets would be missed due to drift, but would also allocate data volume efficiently, was warranted. Rather than choosing from 52 second observations to add to the schedule, we adapted CLASP to schedule from the second a target was predicted to be observed, and then build the observations from there accounting for some amount of uncertainty in the position and the fixed observation size.

The new method of scheduling observations is outlined in Algorithm 2. When a target is attempted to be scheduled that is visible at time $t$, we create an observation record that holds the start time ($st$), end time ($et$), as well as the latest start time ($lst$) and earliest end time ($eet$). These last two parameters are necessary when merging observations. We have two time-dependent functions $p_b$ and $p_a$, which determine the amount of pad time necessary for a target visible at time $t$ to ensure it is not missed. The latest start time and earliest end time will be $t - p_b(t)$ and $t + p_a(t)$ respectively. We consider three ways to determine $st$ and $et$ by shifting

```
procedure schedule()
    write resets at week ends
    run clasp with high priority campaigns
    last_reset_point = start_time
    while progress is made do
        last_reset_point =
          findNextReset(last_reset_point)
        write last_reset_point to file
    end
    run clasp with high and low priority campaigns
procedure findNextReset(lower_bound)
    upper_bound = find upper bound based on
      lower_bound
    while reset time not found do
        t = upper_bound
        while t > lower_bound do
            if memory at time t < threshold then
                return t
            else
                decrement t
            end
        end
        increment threshold
        t = upper_bound
    end
```
**Algorithm 1:** Algorithm for scheduling ring buffer resets

the observation forward or backwards. We choose the first shifting strategy, if any, that results in the observation being able to successfully be added to the schedule. We can center the observation, so that the amount of pad time on either side of $lst$ and $eet$ are equal. We can also make the observation as early or as late as possible, by adding all extra time before $lst$ or after $eet$ respectively.

Then we check to see if this observation is interfering with any previously scheduled observations. Interference could be a direct overlap in time spent observing, or it could violate the minimum length necessary between observations. If this observation does not interfere with any previously scheduled observations, and it does not violate any other constraints (memory, keepout times) it can be placed, and any targets observed during $(t, t + 1)$ have one viewing requirement satisfied. If this observation does interfere with surrounding observations that, we merge this observation and the interfering one, and recursively merge until there are no interfering observations. In the merging algorithm, we check for interference between the newly created observation ($x$) and its immediate neighbors. For the preceding neighbor $n_1$, if there is interference, we create a new observation that has

$$lst \leftarrow min(x.lst, n_1.lst)$$

and

$$eet \leftarrow max(x.eet, n_1.eet)$$

This ensures that for any targets satisfied by O or N, the amount of pad time required on either side of them is maintained. Then we extend the observation to a multiple of a

scene by setting $st$ and $et$ using the current shifting strategy. Figure 9 shows an example of created $x'$ from merging $x$ and $n_1$. We then recursively merge with this new observation ($x'$), and check for interference with the following neighbor $n_2$ and recursively merge once again if necessary. Once we obtain the newly merged observation, we can check if it violates any other constraints. If it does not violate any constraints, we can delete from the schedule any old observations that were merged to form this new one, and place the new one in the schedule. If it does, we consider the next shifting strategy for determining $st$ and $et$, and return that the observation is not able to be placed once we consider all three strategies.

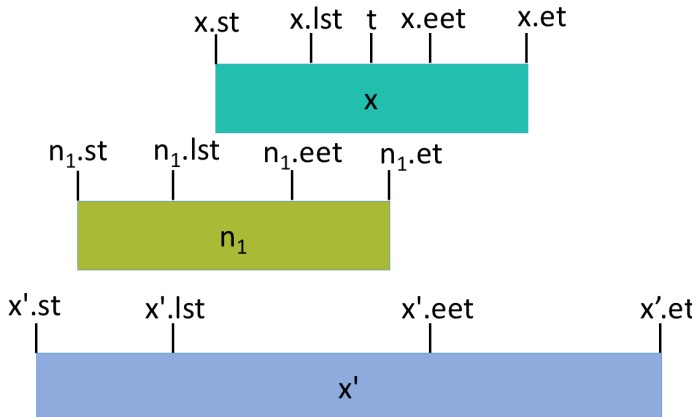

Figure 9: $x'$ is the result of merging the newly created observation $x$ with its preceding neighbor $n_1$

## Validation

We validate the approaches previously presented with two experiments. The first experiment is an analysis of how well the algorithm for scheduling the ring buffer resets performs compared to a schedule produced discounting the issue. The second experiment is a comparison of the schedules produced by the two methods used to account for the uncertainty in the ephemeris.

The algorithm for scheduling the ring buffer resets is analyzed by comparing the schedule produced when accounting for the resets against a schedule produced when we only enforce the data recorder being empty at the end of each week. The goal with the algorithm is to avoid violating Constraints 1 and 2 while still taking as much high priority data as possible.

The change from adding one whole scene to each contiguous set of observations to building observations up from the second each target is observable will be validated by comparing the schedules produced by each method. We use a constant padding function that gives a pad time of 10 seconds on either side of each target. The schedules should have similar numbers of observations, since this value is limited by data constraints, but the resulting coverage should increase with the second method since the data availability is being used more effectively.

```
procedure canPlaceObservation(t)
    for shift in shift strategies do
        create observation x with lst = t − p_b(t) and
            eet = t + p_a(t), st and et according to shift
        x' = mergeObservations(x, shift)
        temporarily delete any interfering observations
            merged to create x'
        if x' does not violate any other constraints then
            put back any deleted observations
            return True
        end
        put back any deleted observations
    end
    return False
procedure mergeObservations(O, shift)
    if x interferes with previous observation n_1 then
        x' = merge x and n_1 together according to
            shift
        x = mergeObservations(x', shift)
    end
    if x interferes with next observation n_2 then
        x' = merge x and n_2 together according to
            shift
        x = mergeObservations(x', shift)
    end
    return x
```

**Algorithm 2:** Algorithm for checking if observations can be placed

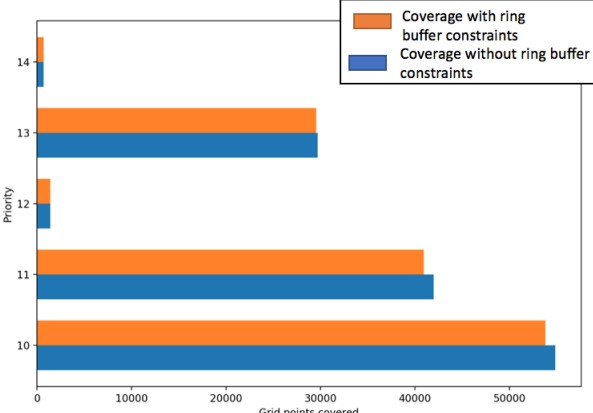

Figure 10: Plot showing gridpoints covered at each priority level by schedules produced when considering or not considering Constraints 1 and 2

scenes from a smaller time delta to adding a scene to each contiguous set of scenes. This shows there had been a significant amount of data volume being wasted with the original padding method. With the original method, the minimum acquisition length was two scenes. In the absolute worst case, if all targets were far enough apart that there were no contiguous scenes, the first method of padding would require double the scenes required by the second method of padding.

## Results

### Ring Buffer Management

Figure 10 shows the coverage amount at each priority level for a schedule produced when considering the ring buffer constraint (orange), and one produced without considering the constraint (blue). When adding in this additional constraint, we achieve a level of coverage of high priority data that is very close to what we would achieve if this was not an issue.

The minimal impact on acquiring the high priority data is due to factors such as the locations of the high priority targets, the instrument data rate, and the downlink rate. There exists times in the schedule with only high priority targets when all of the data has been downlinked and the buffer is empty, allowing resets to be placed with no negative impacts. Had the downlink rate been slower, the instrument data rate been higher, or if there were more high priority targets, it is possible there would be no time when the buffer would be empty.

### Uncertain Ephemeris

Figure 11 shows the difference in target coverage when using the method of building up observations from a smaller time, and Figure 12 shows the number of observations scheduled. For the six weeks tested, there was an average of 29.9% increase in coverage and a 3.75% decrease in observations scheduled when comparing the method of building

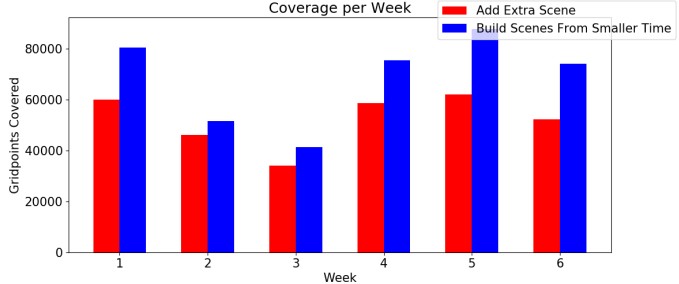

Figure 11: Plot showing coverage difference between the two padding methods

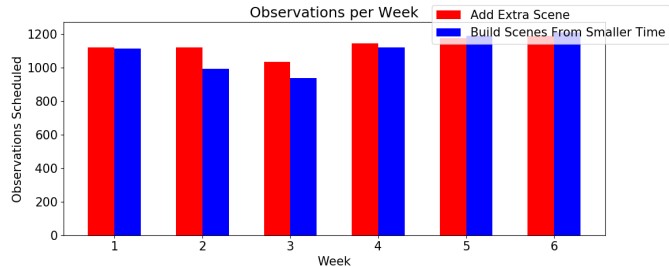

Figure 12: Plot showing number of observations scheduled with the two padding methods

## Related Work

CLASP was previously used for on-orbit scheduling of the IPEX CubeSat (Chien et al. 2015), but IPEX did not require the scheduler to be aware of previously executed schedule or long-term observational campaigns. This paper describes extensions to CLASP that are aware of prior execution and long term observational campaigns.

CLASP has been used for long term mission studies for the upcoming Europa Clipper and JUICE missions (Troesch, Chien, and Ferguson 2017), as well as the NISAR mission (Doubleday and Knight 2014). The ARIEL mission study (Roussel et al. 2017) also focused on long term observation planning. The ARIEL and Europa Clipper studies assume perfect knowledge of future ephemeris and certain execution of scheduled observations, which is appropriate for early mission design analysis, but not mission operations. This paper focuses on the mission operations use case, where the schedule is continuously updated to handle missed/unsatisfactory observations and changes in the observer's ephemeris.

CLASP was also used as a prototype for early stage mission planning of the THEMIS instrument on the Mars Odyssey spacecraft (Rabideau et al. 2010). The focus in the THEMIS study is performance of the squeaky wheel scheduling algorithm. This paper only considers a single pass of squeaky wheel when scheduling.

The receding horizon, sliding window scheduling approach has been implemented for Earth observational scheduling before (Lemaître et al. 2002; Aldinger et al. 2013; Lewellen et al. 2017). These three papers assumed perfect knowledge of vehicle state, perfect execution and focused on optimization and orientation path planning for agile spacecraft. ECOSTRESS is not agile and this paper does not explore optimization. This paper uses the sliding window scheme to address only imperfect state knowledge on longer timescales.

## Future Work

Our decision to require the data recorder to be empty allowed for easier operations because it did not require an interface between the actual vehicle telemetry and the initial conditions of our data recorder fill state model. This simpler interface came at a cost – we prevent the scheduler from taking new science data near the end of each planning period so that the data recorder can drain. ECOSTRESS could produce more science data if we seeded the data recorder fill state with a predicted fill level based on the prior schedule or actual vehicle telemetry. Future missions should consider interfacing data recorder telemetry with the initial conditions of the scheduler's data recorder resource model.

A correctly scheduled and executed observation may be useless because of cloud cover at the time of observation. System malfunctions may also prevent the instrument from executing the scheduled observations. Both of these conditions require an observation to be rescheduled. Future work should handle the previous week's schedule carefully, removing activities that were not actually executed and preserving the resource consumption, but removing the the

credit of unsatisfactory observations.

Currently a constant padding function is used when deciding the earliest end and latest start times. This value is an upper bound on the amount of drift there may be in a one week period. The drift may be time-dependent. The farther an observation from the creation of the ephemeris, the more likely the drift is larger. A better understanding of the drift may allow the padding functions to be truly time-dependent and allow for more observations to be scheduled.

## Conclusion

This paper has described the use of an automated scheduling system in the analysis and operations for the ECOSTRESS mission. Changing orbital ephemeris and long-term campaign goals required adapting CLASP to consider past observations in scheduling for the future. The issue with the instrument ring buffer required scheduling with additional constraints, as well as scheduling another type of instrument activity. The uncertainty of the ISS orbital position required adapting how observations are scheduled. Through computational analysis we showed that our method for addressing the ring buffer approached the performance of schedules produced that did not have the added constraints, and that the second method of building observations up rather outperformed the method of adding a fixed amount of observational time to ensure no regions of interest were missed.

## Acknowledgements

This work was performed at the Jet Propulsion Laboratory, California Institute of Technology, under a contract with the National Aeronautics and Space Administration.

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
