# OpenReview forum: "Automated Science Scheduling for the ECOSTRESS Mission"
_icaps-conference.org/ICAPS/2019/Workshop/SPARK — SPARK 2019_

### Official Review · AnonReviewer1 · 2019-05-01
**An interesting paper**

**Rating:** 4
**Confidence:** 2

**Review:**

The paper presents a scheduling method for satellite observations, specifically for NASA's ECOSTRESS mission aiming at measuring how much water plants need and how plants respond to stress. In particular, the algorithm is an adaptation of the CLASP system that has been used for the same purpose. The results indicate good performance of the proposed method as the number of observations as well as coverage increased.

The problem is well specified and motivated. I like the part that discusses uncertainty as it is one of the important aspect to deal with when a plan, or a schedule has to be executed/applied in real world.

My only critical comment concerns results and their (lack of) discussion. For example, Figure 10 shows that enforcing "ring buffer" constraints has a little effect to performance and the text just summarized the figure. I would expect at least a brief discussion about why the "ring buffer" constraints have such a little performance effect. Analogously, the results of the second experiment could have been explained in more detail.

The Discussion section is rather unclear as it refers to a bug, which is mentioned earlier in the text. I understand that fixing the bug onboard might be too risky and hence updating the software of the ground is a viable option. However, the sentence "This experience has shown us that planning and scheduling work isn't finished at launch" is confusing and perhaps needs a bit more elaboration.

---

### Official Review · AnonReviewer2 · 2019-05-06
**An interesting paper on how to handle domain specific issues adapting an existing scheduling system**

**Rating:** 4
**Confidence:** 3

**Review:**

This paper presents an adaptation of an automated scheduling system, CLASP, to target an EO experiment (ECOSTRESS) on the ISS.

This particular problem presents some peculiarities wrt more standard observation scheduling problems for similar scenarios that required an adaptation of the scheduling system:

- a long-term science campaigns requiring to be able to account for previously executed observations when scheduling for the future
- an issue with the instrument on-board memory required a specific memory management approach
- the uncertainty in the orbital ephemeris requires scheduling flexibility to to ensure no targets are missed at execution time (schedules in this scenario are pre-computed and uploaded for execution)

The topic is interesting and it fits the SPARK workshop.

The paper is clearly written and good balanced between the scenario description, the problem definition and the technical approach presentation. The application is mature and in use.

The solution presented appears interesting, especially the one to tackle into account the memory management issue (long-term science campaigns problems and uncertainty on observations' outcomes are let's say more "standard" issues). Degradation of assets to during their life-cycle is definitely a problem in space applications, and this paper presents an interesting solution.

One things wasn't clear to me: what is exactly a "buffer for contingencies" mentioned when describing the sliding windows approach?

---

### Public Comment · ~Christophe_GUETTIER1 · 2019-05-01
**General comments**

This paper is very clear, the application well presented, and the methodology is credible and mature. The ringbuffer technique, driven by a constraint model, is elegant. On the overall, a good example of autonomous system engineering applied to earth observation. The state of the art is also well addressed with respect to optimisation techniques versus perfect knowledge. However, a comparison with a full constraint solving approach would be interesting.

A question: why mathematical integrals over real have been choosen?

---

### Decision · Program_Chairs · 2019-05-08
**Acceptance Decision**

**Decision:**

Accept

**Comment:**

A welcome submission to SPARK.